# High level of HIV false positives using EIA-based algorithm in survey: Importance of confirmatory testing

Ângelo do Rosário Augusto[1☯], Nnaemeka C. Iriemenam[2☯]*, Luciana Kohatsu[2,3], Leonardo de Sousa[3], Cremildo Maueia[1], Christine Hara[3], Flora Mula[1], Gercio Cuamba[1], Imelda Chelene[1], Zainabo Langa[1], Nathaniel Lohman[4], Flavio Faife[3], Denise Giles[3], Acacio Jose Sabonete[1], Eduardo Samo Gudo[1], Ilesh Jani[1], Bharat S. Parekh[2]*

1 Instituto Nacional de Saúde (INS), EN1, Distrito de Marracuene, Maputo, Mozambique, 2 International Laboratory Branch, Division of Global HIV and TB, Center for Global Health, Centers for Disease Control and Prevention, Atlanta, Georgia, United States of America, 3 Centers for Disease Control and Prevention, Av Zedequias Manganhela, Maputo, Mozambique, 4 The United States Agency for International Development (USAID), Maputo, Mozambique

☯ These authors contributed equally to this work.
* iqd4@cdc.gov (NCI); bsp1@cdc.gov (BSP)

**Data Availability Statement:** All relevant data are within the manuscript.

**Funding:** This Mozambique National Immunization, Malaria and HIV Indicator Survey, 2014 (IMASIDA)

## Abstract

The Mozambique Indicators of Immunization, Malaria and HIV/AIDS (IMASIDA) survey was conducted in 2015 and used a two Enzyme Immunoassay (EIA) (Vironostika HIV-1/2 and Murex HIV-1/2) based algorithm to determine the HIV status of the consented participants. The Mozambique Ministry of Health, with support from the US Centers for Disease Control and Prevention (US CDC), added Bio-Rad Geenius™ HIV-1/2 Supplemental Assay to the IMASIDA HIV testing algorithm to confirm all specimens that were found to be reactive on one or both EIAs. In total 11690 specimens were collected to estimate the proportion of HIV positive samples. Results indicate that the proportion of HIV positive samples based on the concordant positive results of two EIA assays was 21.5% (2518/11690). The addition of the Geenius assay to the IMASIDA HIV testing algorithm demonstrated that 792 (31.5%) of 2518 specimens were false-positive and reduced the proportion of HIV positive samples to 14.7% (1722/11690), demonstrating the importance of including a highly specific HIV test to confirm HIV diagnosis. HIV surveys exclusively based on EIA testing algorithm may result in misleading high prevalence results. Our results demonstrate that more specific confirmatory testing should be added to the EIA-based algorithms to ensure accurate HIV diagnosis and correct HIV prevalence estimate in cross-sectional surveys.

## Introduction

Effective implementation of the national Human Immunodeficiency Virus (HIV) programs require monitoring the HIV epidemic trends to identify programmatic successes, challenges, and needed improvements. HIV prevalence estimates, defined as the percentage of a population affected by HIV, are calculated by testing a representative sample in

has been supported by the President's Emergency Plan for AIDS Relief (PEPFAR) through the Centers for Disease Control and Prevention (CDC) under the terms of Strengthening the National Institute of Health in the Republic of Mozambique, CoAg # GH00080. The funders had no role in study design, data collection and analysis, decision to publish, or preparation of the manuscript. Disclaimer The findings and conclusions in this report are those of the authors and do not necessarily represent the official position of the funding agencies.

**Competing interests:** The authors have declared that no competing interests exist.

the national population surveys [1]. Laboratory-based serological testing using Enzyme Immunoassay (EIA), either in serial or parallel algorithm is often used as the gold standard for estimating HIV prevalence in HIV surveys. EIA is a HIV screening test developed to achieve the highest sensitivity at the cost of expected false positive results [2, 3]. Ongoing development of third and fourth generation EIAs with high sensitivity has reduced the seroconversion window period which is the length of time it takes for an infected person to develop specific antibodies but has increased the potential for false positivity. Moreover, poor laboratory practices further contribute to false positive EIA results [4, 5]. Previous studies have documented false positivity of EIA [6–8], which often lead to over-estimation of HIV prevalence. Also, expert reviews indicate that CD5$^+$ and early B-lymphocyte response to polyclonal cross reactivity and/or potential heterophilic antibody interference might cause false HIV positivity [9, 10]. Therefore, additional supplemental testing using more specific tests such as Western blot or Geenius has been part of the testing algorithm for HIV diagnosis in most Western countries. However, confirmatory testing is usually not performed before registering any HIV positive result during surveillance [11].

In Mozambique, HIV remains a substantial public health burden. In 2009, the Mozambican HIV prevalence among the general population was estimated to be 11.5% [12], and in 2015, according to Indicators of Immunization, Malaria and HIV/AIDS (IMASIDA) survey, the weighted national HIV prevalence among the general population increased to 13.2% [13]. The 2015 IMASIDA survey focused on key health indicators–HIV and malaria as well as measurement of immunization indicators. The primary goal of IMASIDA was to understand HIV prevalence, incidence, and health risk behaviors of the general household populations in Mozambique as well as malaria parasite prevalence among children 6–59 months. The original 2015 IMASIDA survey protocol used a two HIV-1/2 EIA screening test (Vironostika-HIV-1/2 and Murex HIV-1/2) in a serial algorithm to determine the HIV status of the participants and to estimate National HIV prevalence from the survey. Both Vironostika-HIV-1/2 and Murex HIV-1/2 EIAs are known to be highly sensitive and have the potential for nonspecific reactions [14, 15]. To enhance the accuracy and prevent an overestimation of HIV prevalence, the Mozambique Ministry of Health with support from the International Laboratory Branch of the US Centers for Disease Control and Prevention, amended the 2015 IMASIDA protocol and its HIV testing algorithm. A more specific supplementary assay (Bio-Rad Geenius™ HIV-1/2 Supplemental Assay) was added to the IMASIDA testing algorithm in order to align with the latest Joint United Nations Program on HIV and AIDS (UNAIDS)/World Health Organization (WHO) guidance in linked surveys [16]. Bio-Rad Geenius™ HIV-1/2 Supplemental Assay is Food and Drug Administration (FDA) approved and has been evaluated for the confirmation of HIV infection [17]. Geenius, when compared with other HIV confirmatory assays, is less complex, has shorter assay time, can differentiate HIV-1 from HIV-2 antibodies, and has an automated Geenius reader eliminating subjectivity of interpretation [17, 18].

The testing algorithm modification would allow to differentiate HIV-1/2 antibodies and would identify true HIV positives [19] from the two-screening EIA testing. Lessons learned from HIV Clinical Trials in Uganda and Russia showed that confirmatory HIV testing could improve accuracy of HIV diagnosis [20]. In this article, we describe the impact of Bio-Rad Geenius^TM HIV-1/2 Supplemental Assay as the confirmatory test to the 2015 IMASIDA HIV Survey Algorithm in improving the accuracy of HIV testing and providing reliable estimate of HIV prevalence in Mozambique.

## Methods

### Survey methods

The 2015 IMASIDA survey was conducted in 7368 residential households selected from 11 provinces of Mozambique. The criteria for inclusion for HIV/AIDS was household residents and guests who spent the night in the household before the survey. Children aged 6–23 months with parental or guardian consent, and women and men aged 15–59 years who consented to participate were selected. The sampling frame was based on the 2007 Mozambique Population and Housing Census [21]. The sample allocation was multi-staged and stratified in two stages. The first stage comprised the enumeration area (EA) selection using probability proportional to the EA population size, and the second stage had fixed households in every cluster using probability systematic sampling. Each of the 11 provinces were divided into urban and rural settings, making a total of 21 sampling strata. In total, 11690 specimens were collected from the general participants and we estimated the proportion of HIV positive samples. The 2015 IMASIDA protocol received ethical approvals from the Comité Nacional de Bioética para Saúde (CNBS) Institutional Review Board (IRB) and ICF Macro IRB. The protocol was also reviewed in accordance with the Centers for Disease Control and Prevention (CDC) human research protection procedures and was determined to be research, but CDC investigators did not interact with human subjects or have access to identifiable data or specimens for research purposes.

### Sample collection and testing with EIAs

The IMASIDA protocol utilized Dried Blood Spots (DBS) for sample collection, as recommended by WHO for HIV surveillance, especially in resource limited settings [16]. After the volunteer participant's blood collection, two barcoded Whatman 903 filter papers were filled with five spots of 75 μl of blood each, air dried on a DBS drying rack, and placed in an airtight individual ziplock bag with desiccants as well as a humidity indicator card. Samples were transported to INS, for extended storage at -75˚C and testing. All stored DBS cards were de-identified. DBS have been shown as alternate to serum and plasma in many sero-epidemiological studies. The EIA were optimized for DBS use in the laboratory based on published method [22]. Specimens from eligible individuals (15–59 years) and children aged 18–23 months were tested with EIAs Vironostika HIV-1/2, Murex HIV-1/2, and Bio-Rad Geenius™ HIV-1/2 Supplemental Assay in the laboratory according to the manufacturer's instructions, respectively [14, 15, 23]. All Geenius cassettes were read and interpreted using the automated Geenius reader. The specimens that were non-reactive on Vironostika HIV Uni-Form II Ag/Ab (A1–) were classified as HIV negative. Specimens that were reactive on Vironostika HIV Uni-Form II Ag/Ab (A1+) were tested with Murex HIV Ag/Ab Combination (Fig 1A). In addition, 5% of the non-reactive on Vironostika EIA per plate were tested on Murex HIV Ag/Ab EIA for quality assurance.

### Supplemental testing with Geenius HIV-1/2

Additionally, DBS specimens that were dually reactive or reactive with either of the two EIAs were tested using Geenius HIV-1/2 supplementary assay [24]. IgG from the sample, including HIV antibody if present, was eluted from a 6mm punch using 75 μl of PBS-Tween buffer following overnight incubation at 4˚C. A 30 μl aliquot from eluted material was then used to perform Geenius assay. Due to inter-reader variability and subjective nature of manual reading, all Geenius cassettes were read using an automated Geenius reader. All Geenius indeterminate (A3 Ind), irrespective of the specimen EIA statuses, were classified as HIV indeterminate.

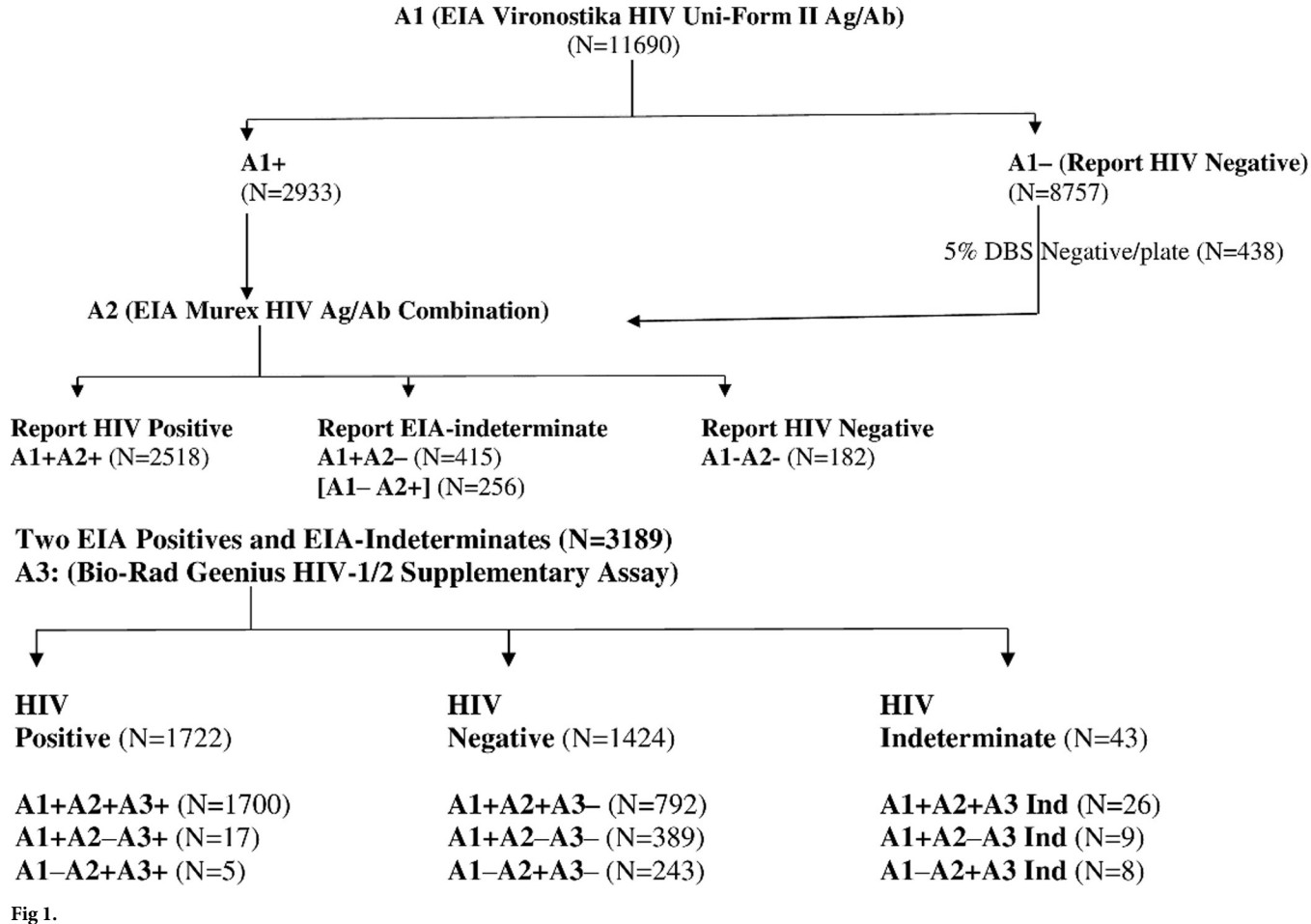

**Fig 1.**

## Statistical analysis

The test results were all entered independently in Microsoft Excel (Microsoft Corporations, Redmond, USA) by two laboratory personnel. All entries were double-checked for errors, duplications, and later merged before performing any data analysis. Analysis was completed using laboratory spreadsheets as the data source. The Pearson chi-square ($\chi^2$) test was used to determine the relationship with categorical variables while Wilcoxon rank sum test was used for continuous variables. Percentage overall agreement was calculated to show the relationship between the dual EIA test results and Geenius final HIV-1/2 classification. In addition, Positive Predictive Value (PPV), with the corresponding 95% Confidence Interval (CI), was calculated to confirm dual EIA HIV positive, using Bio-Rad Geenius™ HIV-1/2 Supplemental Assay as the gold standard. The confidence bounds were calculated using exact binomial bounds.

The tests were two-tailed and P value of $\leq 0.05$ was considered as statistical significance. Data analysis was performed using IBM SPSS Statistics version 21.0 (IBM Corporation, Armonk, NY, USA).

## Results

Fig 1A summarizes the dual EIA flow chart based on the original IMASIDA HIV Surveillance Testing Algorithm. Out of the total 11690 specimens tested using Vironostika HIV-1/2 EIA

(A1), 25.1% (2933) were reactive. All A1 reactive samples (N = 2933) and 5% of negative samples (N = 438) were tested with Murex HIV Ag/Ab Combination EIA (A2). Of the 3371 tested, 2518 (74.7%) were reactive using the A2 while 597 (17.7%) of 3371 were non-reactive on A2 (415 were A1 positive [+], A2 negative [−] and 182 were A1 negative [−], A2 negative [−]). A total of 671 (20.0%) of 3371 were reactive with one of the two EIAs but not by both and therefore were classified as EIA-indeterminate. Based on the dual EIA algorithm, the proportion of HIV positive samples was 21.5% (2518/11690).

## Addition of supplementary Geenius HIV-1/2 assay

Fig 1B details the IMASIDA testing outcome following the addition of Bio-Rad Geenius HIV-1/2 Supplementary Assay (A3). Due to concerns about quality of testing with EIAs, we also included all EIA indeterminates for further testing. Among a total of 3189 specimens tested (2518 EIA-positive and 671 EIA-indeterminates) using Geenius Supplementary Assay, 1722 (54%) were confirmed as HIV positive, while 1424 (44.7%) were Geenius negative and 43 (1.3%) were Geenius indeterminates. Interestingly, 792/2518 (31.5%) of the dual EIA positive specimens were confirmed as HIV negative, and 1% (26) were classified as HIV indeterminate by Geenius HIV-1/2 Supplementary Assay (Fig 1B). Using Geenius supplementary assay results as the final HIV status, PPV of two-EIA-based algorithm was 67.5%, 95% CI [65.6, 69.3] (Table 1).

The percentage overall agreement between the EIA-based algorithm and Geenius HIV-1/2 Supplementary Assay was 53.8%, 95% CI [52.1, 55.6]. Additionally, Geenius assay showed fewer HIV indeterminate results (1.3%, 43/3189) than the EIA indeterminate (21%, 671/3189), and 94.2% of EIA indeterminates (632/671) were confirmed by Geenius assay as HIV negative (Table 1). Furthermore, 3.3% (22/671) of the EIA indeterminates were confirmed as HIV positive by Geenius HIV-1/2 Supplementary Assay summing the proportion of HIV positive samples to 14.7% (1722/11690) [Table 1].

Table 2 shows the Geenius banding patterns of these 22 EIA indeterminate specimens confirmed by Geenius as HIV-positive. Results revealed the three banding patterns demonstrating antibodies to following proteins:

- banding pattern 1: gp160, gp41 which occurred in 23% (5) of the 22 specimens

- banding pattern 2: gp160, p24, gp41 which occurred in 41% (9), and

- banding pattern 3: p31, gp160, p24, gp41 which occurred in 36% (8).

## Signal-to-Cutoff ratios

Fig 2 shows the Signal-to-Cutoff (S/CO) ratios of Vironostika HIV-1/2 EIA and Murex HIV-1/2 EIA with Geenius HIV-1/2 final classification. Compared to the middle quartile values of

**Table 1. Summary of the dual EIA results compared to Geenius supplementary assay results.**

| Final EIA results–A1 and A2 | Geenius Final HIV Status | | | | PPV | Percentage Overall Agreement |
|---|---|---|---|---|---|---|
| | Positive | Negative | Indeterminate | Total | % (95% CI) | (95% CI) |
| Positive | 1700 | 792 | 26 | 2518 | 67.5% (65.6 -69.3%) | 53.8% (52.1%– 55.6%) |
| Negative | 0 | 0 | 0 | 0 | | |
| Indeterminate | 22 | 632 | 17 | 671 | | |
| Total | 1722 | 1424 | 43 | 3189 | | |

Note: PPV = Positive Predictive Value; CI = Confidence Interval.

**Table 2. Geenius banding pattern of the 22 EIA indeterminate samples that are confirmed as HIV positive.**

| Participants | Geenius banding pattern |
|---|---|
| 1 | gp160, gp41, CTRL |
| 2 | gp160, gp41, CTRL |
| 3 | gp160, gp41, CTRL |
| 4 | gp160, gp41, CTRL |
| 5 | gp160, gp41, CTRL |
| 6 | gp160, p24, gp41, CTRL |
| 7 | gp160, p24, gp41, CTRL |
| 8 | gp160, p24, gp41, CTRL |
| 9 | gp160, p24, gp41, CTRL |
| 10 | gp160, p24, gp41, CTRL |
| 11 | gp160, p24, gp41, CTRL |
| 12 | gp160, p24, gp41, CTRL |
| 13 | gp160, p24, gp41, CTRL |
| 14 | gp160, p24, gp41, CTRL |
| 15 | p31, gp160, p24, gp41, CTRL |
| 16 | p31, gp160, p24, gp41, CTRL |
| 17 | p31, gp160, p24, gp41, CTRL |
| 18 | p31, gp160, p24, gp41, CTRL |
| 19 | p31, gp160, p24, gp41, CTRL |
| 20 | p31, gp160, p24, gp41, CTRL |
| 21 | p31, gp160, p24, gp41, CTRL |
| 22 | p31, gp160, p24, gp41, CTRL |

Note: p31 = HIV-1 integrase; gp160 = HIV-1 envelope recombinant protein; p24 = HIV-1 core recombinant protein; gp41 = HIV-1 envelope peptide; CTRL = control line.

the HIV positive S/CO ratio, the middle quartile values of the HIV indeterminate specimens for both Vironostika and Murex EIAs were statistically significantly lower (P < 0.001 and P < 0.001), respectively. Similarly, the middle quartile values of the HIV negative S/CO ratios for both Vironostika and Murex EIAs were statistically significantly lower (P < 0.001 and P < 0.001), respectively than that of the HIV positive S/CO ratio.

## Discussion

This work clearly demonstrates that EIA-based testing algorithm is not sufficient to ensure accuracy of HIV diagnostic results, whether used in survey settings or for individual diagnosis. EIAs are designed to be used as screening assays and are very sensitive for early diagnosis of HIV infections. This is specifically achieved by third generation EIAs that can detect both IgG and IgM and fourth generation EIAs that also can detect p24 antigen, in addition to HIV antibodies, present during acute phase of infection before antibodies develop. However, push to increase sensitivity leads to some level of false reactivity, which is well recognized [25, 26]. WHO recommends an HIV testing strategy combining a very sensitive test with a second more specific test to accurately identify HIV antibodies present in the sample. However, this guidance is often overlooked, and many laboratories have used two or more EIAs (all very sensitive) to perform HIV testing for surveys or for diagnosis. This has led to significant number of common false reactivity resulting in elevated prevalence in surveys and false-positive diagnosis of individuals. In addition to inherent properties of EIAs, poor practices in laboratories

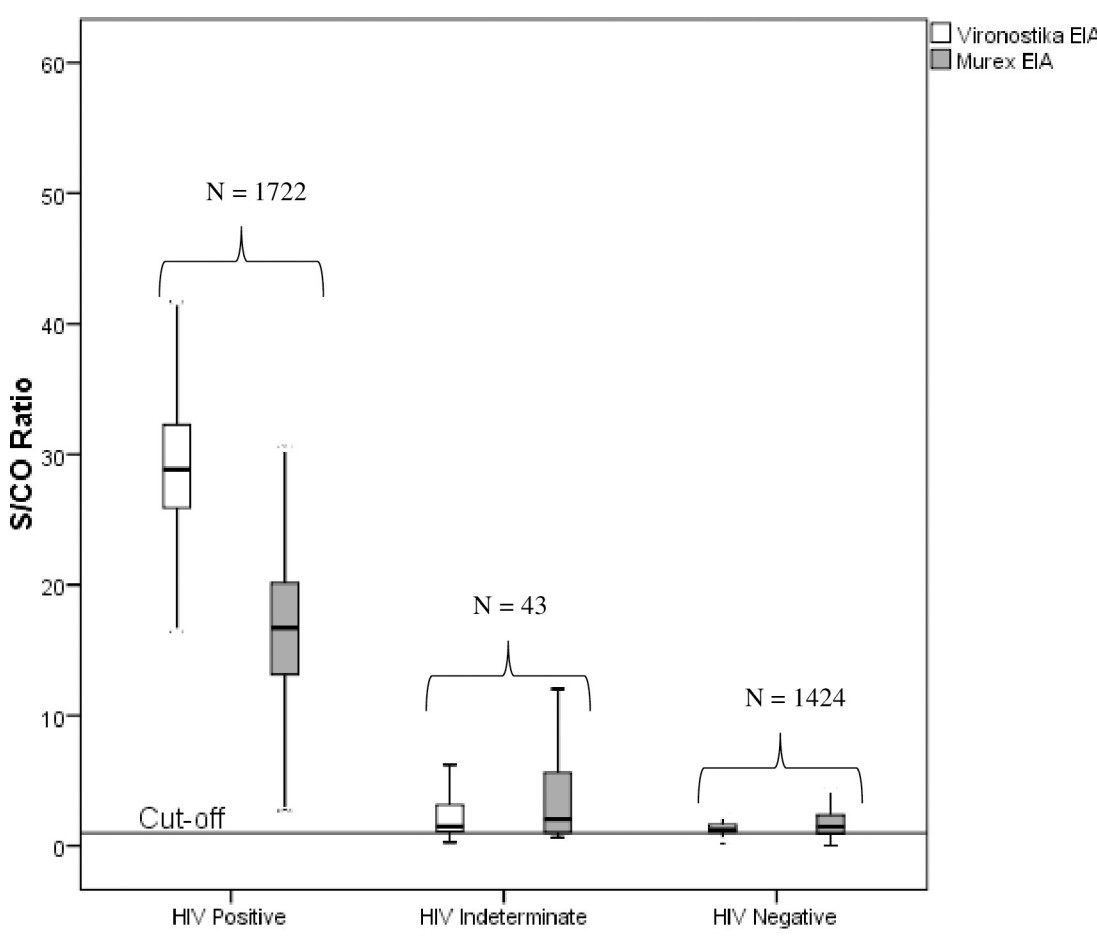

**Fig 2.**

and lack of maintenance of EIA equipment have resulted in high false positives. Some antenatal clinic surveys, demographic health surveys and AIDS indicator surveys routinely used EIA-based testing strategy resulting in overestimation of HIV prevalence in most surveys [27]. Unfortunately, these prevalence data may lead unreliable numbers when used for modeling by UNAIDS and others to extrapolate HIV burden in different countries and populations leading to adverse impact on planning and resource allocation. Often laboratory results are interpreted by epidemiologists or modelers without any input from laboratory experts. Engagement of laboratory experts with right skills in the survey design, protocol development, data collection, data review, and interpretation during the survey is critical to ensure results are reliable and accurate.

Our results show that IMASIDA survey had 31.5% of individuals that were false positives when two-EIA-based testing algorithm was used. There were 671 EIA-indeterminate specimens, where one of the two EIAs was reactive (Table 1). Further testing by Geenius indicated that only 22/671 (3.3%) were HIV-positive while the remaining were either Geenius negative (94.2%, i.e. 632/671) or indeterminate (2.5%, i.e. 17/671). This demonstrates high level of false reactivity by EIA but also highlights some true positives that were not concordantly positive by two EIA reflecting lack of quality assurance in performing the EIAs. In addition, the banding pattern of these 22 specimens showed presence of distinct antibodies. Antibodies to group-

specific antigen protein p24 is one of the earliest to appear after HIV infection and followed by the envelop glycoprotein gp160 and gp41 predicting early stage of HIV infection [28]. In addition, antibodies to integrase gene p31 appear much later which can predict later stage of HIV infection. These results further confirm poor testing practices highlighting need for additional attention to overall quality assurance.

High sensitivity of EIAs make them perfectly suitable for blood banks to screen blood donations for any potentially HIV-positive units. This is done at the cost of discarding some false-reactive blood units. However, this approach is not suitable for individual diagnosis as well as for surveys. Due to expansion of effective Antiretroviral Therapy (ART), unlinked surveys are not recommended anymore; this means any result of HIV testing should be returned to individuals which further requires high level of accuracy of testing in surveys. The results returned to individuals in the case of IMASIDA was based on the national routine HIV rapid testing algorithm rather than the laboratory-based EIA testing. Based on the two EIA-testing algorithm, the proportion of HIV positive samples would have been 21.5%. However, upon the introduction of Geenius HIV-1/2 supplementary assay testing, this proportion of HIV positive samples reduced to 14.7%. Remarkably, 31.5% of the dual-EIA positive participants were false positive. Recognizing that 4th generation EIAs can detect p24 antigen present during acute infections, additional nucleic acid testing (NAT) was performed on Geenius negative/indeterminate specimens. This identified only 11 (1.4%) NAT-positive acute infections further confirming that most specimens were false positive. The introduction of Geenius assay to the IMASIDA algorithm demonstrates that the prevalence data based on EIA testing alone may result in misleading high prevalence results and emphasizes the importance of including a highly specific HIV test to HIV surveillance testing protocols. Our results confirm that it is crucial to include a HIV confirmatory assay to cross-sectional surveys that are based on single or dual EIA algorithm. DBS testing using immunoassay has been shown as a dependable alternate for HIV testing as well as surveillance [29]. However, optimization cannot eliminate all false positives that are inherent to EIAs. The utility of Geenius HIV confirmation using DBS specimen has been established and proven as a rapid as well as a reliable alternate to other HIV confirmation assays [24]. Additionally, the performance of Geenius HIV-1/2 supplementary assay has similarly been documented in multiple public health testing populations [30–33].

The causes of false HIV positives in enzyme immunoassays may be multi-factorial [34]. For example, it could be due to administrative error, lack of competently trained laboratory staff, poor quality control, lack of assay optimization with DBS, failure to maintain equipment, poor assay specificity, cross reactivity, and interference from heterophilic antibodies. Fourth generation EIAs (like Vironostika Ag/Ab and Murex Ag/Ab) detect HIV-1 and HIV-2 antibodies and p24 antigen with high sensitivity [34]. Fourth generation EIAs offers better advantages over other generations of EIA but poses the risk of high rate of false reactivity with low PPV. Previous study indicated that increasing the S/CO ratio might rule out false positive results by increasing its positivity threshold [35]. Nevertheless, raising the S/CO ratio to increase specificity might as well result in elevated false negative results [36]. In our study, the median S/CO ratios of Vironostika and Murex for the Geenius-confirmed HIV-positive samples were significantly higher than the median S/CO ratios of Vironostika and Murex for the Geenius-confirmed false reactive samples.

In addition, when all nonspecific factors leading to false positive results are controlled (e.g., administrative error, equipment maintenance, assay optimization, competently trained laboratory staff, poor quality control), heterophilic antibody interaction remains a concern. Future research on the interference of heterophilic antibodies is warranted given that it can be an important cause of false positive HIV results in enzyme immunoassays [37, 38]. The Geenius™ HIV-1/2 Supplementary Assay is an FDA approved HIV-1/2 confirmatory assay and

has the option of an automated traceable reading removing any risk of human error from the laboratory staff in the interpretation of results. In our results, the Geenius HIV-1/2 confirmatory assay showed fewer indeterminate results than the EIA indeterminate, and confirmed 3.3% of the EIA indeterminate as HIV positive as well as 67.5% of the dual-EIA HIV positive. Due to the socio-economical, cultural, psychological and public health significance of HIV diagnosis, the importance of HIV confirmatory assay is fundamental in order to reduce the risk of false positive test results. In many situations, a positive HIV diagnosis may lead to stigmatization, depression, and suicidal intent of the client [39]. WHO in 2015 recommended ART for all HIV positive individuals [40], and with the universal test and treat, a HIV confirmatory assay using a different method like Geenius™ HIV-1/2 Supplementary Assay, Innolia™ HIV-I/II or HIV-1 Western Blot Assay would be ideal before reporting any HIV positive result. It is to be noted that HIV rapid test-based testing algorithms are more specific and had PPV of 99% or greater, when performed as recommended (unpublished data from Population-based HIV Impact Assessment Surveys).

In summary, the introduction of Geenius confirmatory assay to the IMASIDA HIV testing algorithm underscored the importance of including a more specific HIV-1/2 test during HIV surveillance testing to improve the accuracy and reliability of the HIV test result. Our data shows that HIV survey based only on EIA testing algorithm may result in misleading high HIV prevalence.

## Acknowledgments

The authors would like to thank all IMASIDA 2015 participants that consented to be tested. The authors also thank Keydra Olapado for her technical support throughout the survey. Additionally, the authors thank INS staff for their immense support during the Geenius HIV-1/2 confirmatory assay training and testing of IMASIDA specimens, as well as Artur Ramos for his directorial country engagement.

## Author Contributions

**Conceptualization:** Nnaemeka C. Iriemenam, Bharat S. Parekh.

**Data curation:** Ângelo do Rosário Augusto, Nnaemeka C. Iriemenam, Cremildo Maueia, Christine Hara, Nathaniel Lohman.

**Formal analysis:** Nnaemeka C. Iriemenam, Bharat S. Parekh.

**Funding acquisition:** Luciana Kohatsu, Leonardo de Sousa, Christine Hara, Nathaniel Lohman, Denise Giles, Acacio Jose Sabonete, Eduardo Samo Gudo, Ilesh Jani.

**Investigation:** Ângelo do Rosário Augusto, Nnaemeka C. Iriemenam, Luciana Kohatsu, Leonardo de Sousa, Cremildo Maueia, Flora Mula, Gercio Cuamba, Imelda Chelene, Zainabo Langa, Nathaniel Lohman, Flavio Faife, Acacio Jose Sabonete, Eduardo Samo Gudo, Ilesh Jani, Bharat S. Parekh.

**Methodology:** Ângelo do Rosário Augusto, Nnaemeka C. Iriemenam, Luciana Kohatsu, Leonardo de Sousa, Cremildo Maueia, Christine Hara, Flora Mula, Gercio Cuamba, Imelda Chelene, Zainabo Langa, Nathaniel Lohman, Flavio Faife, Acacio Jose Sabonete, Eduardo Samo Gudo, Ilesh Jani, Bharat S. Parekh.

**Project administration:** Ângelo do Rosário Augusto, Nnaemeka C. Iriemenam, Luciana Kohatsu, Leonardo de Sousa, Christine Hara, Nathaniel Lohman, Denise Giles, Eduardo Samo Gudo, Ilesh Jani.

**Resources:** Nnaemeka C. Iriemenam, Luciana Kohatsu, Leonardo de Sousa, Cremildo Maueia, Flora Mula, Gercio Cuamba, Imelda Chelene, Zainabo Langa, Nathaniel Lohman, Flavio Faife, Denise Giles, Acacio Jose Sabonete, Eduardo Samo Gudo, Ilesh Jani, Bharat S. Parekh.

**Supervision:** Nnaemeka C. Iriemenam, Luciana Kohatsu, Leonardo de Sousa, Christine Hara, Denise Giles, Eduardo Samo Gudo, Ilesh Jani, Bharat S. Parekh.

**Validation:** Nnaemeka C. Iriemenam, Luciana Kohatsu, Cremildo Maueia, Nathaniel Lohman, Flavio Faife, Acacio Jose Sabonete, Ilesh Jani, Bharat S. Parekh.

**Visualization:** Flora Mula, Gercio Cuamba, Imelda Chelene, Zainabo Langa, Nathaniel Lohman, Flavio Faife, Denise Giles, Acacio Jose Sabonete, Eduardo Samo Gudo, Ilesh Jani.

**Writing – original draft:** Nnaemeka C. Iriemenam.

**Writing – review & editing:** Ângelo do Rosário Augusto, Nnaemeka C. Iriemenam, Luciana Kohatsu, Leonardo de Sousa, Cremildo Maueia, Christine Hara, Flora Mula, Gercio Cuamba, Imelda Chelene, Zainabo Langa, Nathaniel Lohman, Flavio Faife, Denise Giles, Acacio Jose Sabonete, Eduardo Samo Gudo, Ilesh Jani, Bharat S. Parekh.

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
