## [Decision Letter · Decision Letter 0]

8 Jul 2020

PONE-D-20-15604

High Level of HIV False Positives using EIA-Based Algorithm in Survey: Importance of Confirmatory Testing

PLOS ONE

Dear Dr. Irlemenam,

Thank you for submitting your manuscript to PLOS ONE. After careful consideration, we feel that it has merit but does not fully meet PLOS ONE’s publication criteria as it currently stands. Therefore, we invite you to submit a revised version of the manuscript that addresses the points raised during the review process.

Your manuscript has been reviewed by two experts in the field. Please fully address the comments and issues raised by the reviewers and resubmit the manuscript for consideration for publication.

We look forward to receiving your revised manuscript.

Kind regards,

Sanjai Kumar

Academic Editor

PLOS ONE

Journal Requirements:

4. Please upload a new copy of Figure xxxx as the detail is not clear. Please follow the link for more information: https://blogs.plos.org/plos/2019/06/looking-good-tips-for-creating-your-plos-figures-graphics/" https://blogs.plos.org/plos/2019/06/looking-good-tips-for-creating-your-plos-figures-graphics/

Reviewers' comments:

Reviewer's Responses to Questions

**Comments to the Author**

1. Is the manuscript technically sound, and do the data support the conclusions?

Reviewer #1: Yes

Reviewer #2: Partly

2. Has the statistical analysis been performed appropriately and rigorously? 

Reviewer #1: Yes

Reviewer #2: Yes

3. Have the authors made all data underlying the findings in their manuscript fully available?

Reviewer #1: No

Reviewer #2: Yes

4. Is the manuscript presented in an intelligible fashion and written in standard English?

Reviewer #1: Yes

Reviewer #2: Yes

5. Review Comments to the Author

Reviewer #1: Nnaemeka Iriemenam et al presented ‘High Level of HIV False Positives using EIA Based Algorithm in Survey: Importance of Confirmatory Testing’. The main focus of this study is assessing effectiveness of HIV prevalence surveys using two Enzyme Immunoassay (EIA) (Vironostika HIV-1/2 and Murex HIV-1/2) from the samples collected in dried blood spots. Study was conducted in 2015 through the program Mozambique Indicators of Immunization, Malaria and HIV/AIDS (IMASIDA) from

11 provinces of Mozambique. About 11690 specimens were collected from the general participants. In original survey protocol two HIV-1/2 EIA screening test (Vironostika-HIV-1/2 and Murex HIV-1/2) was used later the Mozambique Ministry of Health with support from the International Laboratory Branch of the US Centers for Disease Control and Prevention, amended a more specific supplementary assay (Bio-Rad Geenius™ HIV-1/2 Supplemental Assay) to the 2015 IMASIDA protocol and its HIV testing algorithm.

Results indicate that the proportion of HIV positive samples based on the concordant positive results of two EIA assays was 21.5% (2518/11690). The addition of the Geenius assay to the IMASIDA HIV testing algorithm demonstrated that 792 (31.5%) of 2518 specimens were false positive and reduced the proportion of HIV positive samples to 14.7% (1722/11690). Authors claim the introduction of a highly specific HIV test (The Geenius Assay) substantially improves the HIV/AIDS surveillance sensitivity.

This report is very important for the researchers/program administrators involved in HIV/AIDS surveillance. Although this report brings out new information for resource limited setting following issues needs to be addressed before publication.

Issues:

1. It is unclear whether EIA assays used in this study were originally approved for testing with a Dried Blood Spot (DBS) sample matrix. If assay indications for use support this sample matrix, it should be mentioned in the methods section. If EIA assays used are not approved for DBS sample matrix, any previous validation (sensitivity/specificity) at resource limited settings needs to be cited. Also, if data available for serum/plasma sample matrix used for HIV surveillance program at Mozambique it would be useful to mention.

2. In results section, line 6, line starts with “(415 were A1+A2- and 182 were A1-A2-)” It is confusing for readers. Consider revising 415 were A1 positive, A2 negative instead of ‘+’ or ‘-‘ signs.

Reviewer #2: The manuscript “High Level of HIV False Positives using EIA-Based Algorithm in Survey: Importance of Confirmatory Testing” by Ângelo do Rosário Augusto, et al describes the need to add a more specific confirmatory test to the EIA-based algorithms currently being used for the accurate estimation of HIV prevalence. The data provided in this manuscript is of interest to researchers in this field. The manuscript is well written and easy to follow.

The authors should address the following points:

• All data is derived from dried blood spots (DBS). The performance characteristics of all 3 assays used (Vironostika HIV Uni-Form II Ag/Ab assay, Murex HIV-1/2 assay and the Bio-Rad Geenius HIV-1/2 Supplemental Assay) were established using serum or plasma. DBS is not the recommended sample matrix for use with these test kits. The authors have not included any discussion about the sensitivity and specificity of these tests when DBS is used as the sample matrix when compared to the use of serum/plasma.

• The authors should discuss the potential impact of the off-label use of DBS on the study results.

• The Vironostika HIV Uni-Form II Ag/Ab assay and the Murex HIV-1/2 assay both detect HIV-1 p24 antigen, However, the Bio-Rad Geenius HIV-1/2 Supplemental Assay detects antibodies to HIV-1 and HIV-2 only and does not detect HIV-1 p24 antigen. Therefore, if a sample was reactive to HIV-1 p24 only (window period sample) the Bio-Rad Geenius assay will not be reactive. The authors have not included data from additional NAT testing or follow-up testing. The authors should describe how discordant test results were resolved in the absence of additional NAT testing or follow-up testing.

• The performance characteristics regarding the Vironostika HIV Uni-Form II Ag/Ab assay, Murex HIV-1/2 assay and the Bio-Rad Geenius HIV-1/2 Supplemental Assay should be provided. The package insert of these tests will help the reader to better understand the results.

6. PLOS authors have the option to publish the peer review history of their article (what does this mean?). If published, this will include your full peer review and any attached files.

Reviewer #1: No

Reviewer #2: No

---

## [Author Response · Author response to Decision Letter 0]

9 Sep 2020

Response to Reviewers

• Thank you so much for the review of our manuscript. The authors have revised the manuscript according to the reviewers’ comments. We have as well responded point by point to the comments/questions from the reviewers below.

Reviewer #1: 

Nnaemeka Iriemenam et al presented ‘High Level of HIV False Positives using EIA Based Algorithm in Survey: Importance of Confirmatory Testing’. The main focus of this study is assessing effectiveness of HIV prevalence surveys using two Enzyme Immunoassay (EIA) (Vironostika HIV-1/2 and Murex HIV-1/2) from the samples collected in dried blood spots. Study was conducted in 2015 through the program Mozambique Indicators of Immunization, Malaria and HIV/AIDS (IMASIDA) from

11 provinces of Mozambique. About 11690 specimens were collected from the general participants. In original survey protocol two HIV-1/2 EIA screening test (Vironostika-HIV-1/2 and Murex HIV-1/2) was used later the Mozambique Ministry of Health with support from the International Laboratory Branch of the US Centers for Disease Control and Prevention, amended a more specific supplementary assay (Bio-Rad Geenius™ HIV-1/2 Supplemental Assay) to the 2015 IMASIDA protocol and its HIV testing algorithm.

Results indicate that the proportion of HIV positive samples based on the concordant positive results of two EIA assays was 21.5% (2518/11690). The addition of the Geenius assay to the IMASIDA HIV testing algorithm demonstrated that 792 (31.5%) of 2518 specimens were false positive and reduced the proportion of HIV positive samples to 14.7% (1722/11690). Authors claim the introduction of a highly specific HIV test (The Geenius Assay) substantially improves the HIV/AIDS surveillance sensitivity.

This report is very important for the researchers/program administrators involved in HIV/AIDS surveillance. Although this report brings out new information for resource limited setting following issues needs to be addressed before publication.

Issues:

1. It is unclear whether EIA assays used in this study were originally approved for testing with a Dried Blood Spot (DBS) sample matrix. If assay indications for use support this sample matrix, it should be mentioned in the methods section. If EIA assays used are not approved for DBS sample matrix, any previous validation (sensitivity/specificity) at resource limited settings needs to be cited. Also, if data available for serum/plasma sample matrix used for HIV surveillance program at Mozambique it would be useful to mention.

• Response: These assays were optimized for DBS use in the laboratory using published methodology, which is a common practice, specifically for surveys. However, this manuscript highlights multiple factors that are critical in addition to performance of the tests. High level of discordant results between two EIAs in order to get accurate result, a more specific supplementary assay was introduced to the IMASIDA HIV testing algorithm. DBS collected on filter papers have been used as alternative specimens to serum/plasma collection either to increase access to HIV testing at the individual level or to perform sero-epidemiological studies at the population level.

o Chee Eng Lee, S. S. P., Sharifah Faridah Syed Omar, Sanjiv Mahadeva, Lai Yee Ong and Adeeba Kamarulzaman (2011). "Evaluation of the Dried Blood Spot (DBS) Collection Method as a Tool for Detection of HIV Ag/Ab, HBsAg, anti-HBs and anti-HCV in a Malaysian Tertiary Referral Hospital." Ann Acad Med Singapore 40(10): 448-453.

o Stefic, K., J. Guinard, G. Peytavin, L. Saboni, C. Sommen, C. Sauvage, F. Lot, S. Laperche, A. Velter and F. Barin (2019). "Screening for Human Immunodeficiency Virus Infection by Use of a Fourth-Generation Antigen/Antibody Assay and Dried Blood Spots: In-Depth Analysis of Sensitivity and Performance Assessment in a Cross-Sectional Study." J Clin Microbiol 58(1).

2. In results section, line 6, line starts with “(415 were A1+A2- and 182 were A1-A2-)” It is confusing for readers. Consider revising 415 were A1 positive, A2 negative instead of ‘+’ or ‘-‘ signs.

• Response: Thanks for your suggestion. The authors have changed “(415 were A1+A2- and 182 were A1-A2-)” to (415 were A1 positive, A2 negative and 182 were A1 negative, A2 negative) as recommended.

Reviewer #2: The manuscript “High Level of HIV False Positives using EIA-Based Algorithm in Survey: Importance of Confirmatory Testing” by Ângelo do Rosário Augusto, et al describes the need to add a more specific confirmatory test to the EIA-based algorithms currently being used for the accurate estimation of HIV prevalence. The data provided in this manuscript is of interest to researchers in this field. The manuscript is well written and easy to follow.

The authors should address the following points:

• All data is derived from dried blood spots (DBS). The performance characteristics of all 3 assays used (Vironostika HIV Uni-Form II Ag/Ab assay, Murex HIV-1/2 assay and the Bio-Rad Geenius HIV-1/2 Supplemental Assay) were established using serum or plasma. DBS is not the recommended sample matrix for use with these test kits. The authors have not included any discussion about the sensitivity and specificity of these tests when DBS is used as the sample matrix when compared to the use of serum/plasma.

• Response: Based on the kit inserts, the diagnostic sensitivity of Vironostika HIV Ag/Ab is estimated to be 100% and specificity 99.5%. Murex HIV Ag/b combination sensitivity is estimated to be 100% and specificity 99.78%. These assays were optimized for DBS use in the laboratory. However, in order to get accurate result, a more specific supplementary assay was introduced to the IMASIDA HIV testing algorithm. DBS have been shown as alternate to serum/plasma in many sero-epidemiological studies at the population level. 

o Chee Eng Lee, S. S. P., Sharifah Faridah Syed Omar, Sanjiv Mahadeva, Lai Yee Ong and Adeeba Kamarulzaman (2011). "Evaluation of the Dried Blood Spot (DBS) Collection Method as a Tool for Detection of HIV Ag/Ab, HBsAg, anti-HBs and anti-HCV in a Malaysian Tertiary Referral Hospital." Ann Acad Med Singapore 40(10): 448-453.

• For Geenius supplementary assay, the utilization using DBS has been documented and published. This was used as recommended and cited in the manuscript Reference 23 – “Fernandez McPhee C, Alvarez P, Prieto L, Obiang J, Avedillo P, Vargas A, et al. HIV-1 infection using dried blood spots can be confirmed by Bio-Rad Geenius HIV 1/2 confirmatory assay. J Clin Virol. 2015;63:66-9”.

• The authors should discuss the potential impact of the off-label use of DBS on the study results.

• Response: Thank you. The authors have included the off-label use of DBS in the manuscript. However, the tests were optimized to be used with DBS as stated earlier. 

• The Vironostika HIV Uni-Form II Ag/Ab assay and the Murex HIV-1/2 assay both detect HIV-1 p24 antigen, However, the Bio-Rad Geenius HIV-1/2 Supplemental Assay detects antibodies to HIV-1 and HIV-2 only and does not detect HIV-1 p24 antigen. Therefore, if a sample was reactive to HIV-1 p24 only (window period sample) the Bio-Rad Geenius assay will not be reactive. The authors have not included data from additional NAT testing or follow-up testing. The authors should describe how discordant test results were resolved in the absence of additional NAT testing or follow-up testing.

• Response: Thanks for your comment. We agree that Geenius is intended to detect antibodies to HIV-1 and HIV-2 proteins. It does not detect acute infection including p24 antigen. Additional NAT testing of the discordant showed that only 1.4% (11) were HIV-1 positive (we added that in the discussion). Therefore, almost all discordant specimens were false positives as confirmed by Geenius. 

• The performance characteristics regarding the Vironostika HIV Uni-Form II Ag/Ab assay, Murex HIV-1/2 assay and the Bio-Rad Geenius HIV-1/2 Supplemental Assay should be provided. The package insert of these tests will help the reader to better understand the results.

• Response: Thank you. The package inserts of all the assays were cited in the manuscript. 

o Reference 14 – Vironostika HIV Uni-Form II Ag/Ab EIA assay

o Reference 15 – Murex HIV-1/2 EIA assay

o Reference 22 – Bio-Rad Geenius HIV-1/2 Supplemental Assay

---

## [Editor Report · Decision Letter 1]

14 Sep 2020

High Level of HIV False Positives using EIA-Based Algorithm in Survey: Importance of Confirmatory Testing

PONE-D-20-15604R1

Dear Dr. Iriemenam,

We’re pleased to inform you that your manuscript has been judged scientifically suitable for publication and will be formally accepted for publication once it meets all outstanding technical requirements.

Kind regards,

Sanjai Kumar

Academic Editor

PLOS ONE
---

## [Editor Report · Acceptance letter]

13 Oct 2020

PONE-D-20-15604R1 

High Level of HIV False Positives using EIA-Based Algorithm in Survey: Importance of Confirmatory Testing

Dear Dr. Iriemenam:

I'm pleased to inform you that your manuscript has been deemed suitable for publication in PLOS ONE. Congratulations! Your manuscript is now with our production department. 

Kind regards, 

on behalf of

Dr. Sanjai Kumar 

Academic Editor

PLOS ONE